# Influence of Polyphenols on Adipose Tissue: Sirtuins as Pivotal Players in the Browning Process

**DOI:** 10.3390/ijms24119276

**Published:** 2023-05-25

**Authors:** Lorenzo Flori, Eugenia Piragine, Jacopo Spezzini, Valentina Citi, Vincenzo Calderone, Alma Martelli

**Affiliations:** 1Department of Pharmacy, University of Pisa, 56126 Pisa, Italy; lorenzo.flori@farm.unipi.it (L.F.); eugenia.piragine@farm.unipi.it (E.P.); jacopo.spezzini@phd.unipi.it (J.S.); valentina.citi@unipi.it (V.C.); vincenzo.calderone@unipi.it (V.C.); 2Interdepartmental Research Center “Nutrafood: Nutraceutica e Alimentazione per la Salute”, University of Pisa, 56126 Pisa, Italy; 3Interdepartmental Research Center “Biology and Pathology of Ageing”, University of Pisa, 56126 Pisa, Italy

**Keywords:** browning, polyphenols, sirtuins, white adipose tissue, brown adipose tissue, SIRT1, weight control

## Abstract

Adipose tissue (AT) can be classified into two different types: (i) white adipose tissue (WAT), which represents the largest amount of total AT, and has the main function of storing fatty acids for energy needs and (ii) brown adipose tissue (BAT), rich in mitochondria and specialized in thermogenesis. Many exogenous stimuli, e.g., cold, exercise or pharmacological/nutraceutical tools, promote the phenotypic change of WAT to a beige phenotype (BeAT), with intermediate characteristics between BAT and WAT; this process is called “browning”. The modulation of AT differentiation towards WAT or BAT, and the phenotypic switch to BeAT, seem to be crucial steps to limit weight gain. Polyphenols are emerging as compounds able to induce browning and thermogenesis processes, potentially via activation of sirtuins. SIRT1 (the most investigated sirtuin) activates a factor involved in mitochondrial biogenesis, peroxisome proliferator-activated receptor γ coactivator 1α (PGC-1α), which, through peroxisome proliferator-activated receptor γ (PPAR-γ) modulation, induces typical genes of BAT and inhibits genes of WAT during the transdifferentiation process in white adipocytes. This review article aims to summarize the current evidence, from pre-clinical studies to clinical trials, on the ability of polyphenols to promote the browning process, with a specific focus on the potential role of sirtuins in the pharmacological/nutraceutical effects of natural compounds.

## 1. Introduction

AT is a dynamic organ characterized by adipocytes, which are cells specialized in storing energy in the form of lipid droplets [1,2]. AT plays a pivotal role in the regulation of energy storage and thermogenesis and influences many physiological processes such as glucose and lipid metabolism, blood pressure homeostasis, inflammatory and immune responses [3]. AT can be classified into two functionally different types, WAT and BAT [4]. WAT accounts for the greater part of the total AT in the human body [5] and white adipocytes are characterized by a large lipid droplet covering most of the cell volume, while possessing few mitochondria [6]. WAT maintains the body’s energy homeostasis by storing energy in the form of triglycerides and by acting as an endocrine organ that secretes cytokines and adipokines [4]. Adipokines such as leptin and adiponectin, in particular, regulate insulin signaling, fatty acid oxidation, and chemoattraction of immune cells [7,8]. In contrast, BAT is characterized by adipocytes containing multiple small lipid droplets, rich in mitochondria and specialized in thermogenesis [9].

## 2. Pathological Conditions of Adipose Tissue

Adipocyte hypertrophy and hyperplasia result in AT expansion in a condition of chronic surplus of nutrient availability. AT hyperplasia that results from the differentiation of adipocyte progenitor cells is considered a healthy mechanism of AT expansion. Conversely, adipocyte hypertrophy is related to a pro-inflammatory phenotype associated with pro-inflammatory cytokines’ production and the infiltration of immune cells [10,11,12]. AT hypertrophy leads to tissue dysfunction, which is characterized by three main processes: inflammation, fibrosis, and impaired angiogenesis [13]. Unhealthy expansion of AT results in hypoxia due to reaching the diffusional limit of oxygen. The stress signal generated by the lack of oxygen promotes angiogenesis and extracellular matrix remodeling but, when the state of hypernutrition persists over time, the expansion of AT limits angiogenesis and leads to hypoxia, inflammation, oxidative stress, apoptosis, fibrosis, and infiltration of immune cells [14,15]. Furthermore, due to the hypoxia during the early stages of AT expansion, adipocytes release pro-inflammatory cytokines such as interleukin-6 (IL-6) and tumor necrosis factor-α (TNF-α) that promote migration of monocytes and their differentiation into pro-inflammatory macrophages [16,17]. The presence of chronic low-grade inflammation, termed “meta-inflammation”, is a feature of the inflammatory profile of dysfunctional AT [18]. Meta-inflammation promotes a phenotypic switch in AT macrophages (the first immune cell population that has been reported to infiltrate the AT) from the anti-inflammatory and immunosuppressive M2 profile to the pro-inflammatory M1 profile [19]. The M1 macrophages surround the dead adipocytes forming a “crown-like” structure and causing the release of triglycerides, previously contained in adipocytes, which further activate the M1 macrophages and promote the secretion of TNF-α, contributing to the chronicization of the inflammatory state [20,21]. Finally, under homeostatic conditions, adipocytes are surrounded by an extracellular matrix (ECM), consisting mainly of collagen, fibrillin, and proteoglycans, which contains growth factors and aids AT expansion and the differentiation of adipocyte progenitor cells, also providing mechanical support [22]. Excessive accumulation of ECM due to the rapid expansion of AT leads to fibrosis [23]. Fibrosis further contributes to adipocyte dysfunction by altering the mechanical structure of the tissue. This event leads to impaired lipolysis, decreased adipokines’ expression and nutrient availability, increased expression of pro-inflammatory cytokines, and reduced angiogenesis [24,25]. Taken together, all these processes represent a risk factor for the onset of obesity, insulin resistance, type 2 diabetes (T2D), cardiovascular diseases, and metabolic disorders [15,20,26].

## 3. Brown Adipose Tissue and Its Implication in Metabolic Processes

WAT is not the only AT phenotype but is complemented by BAT and BeAT. These tissues differ in both structure and function, but the molecular pathways involved in the regulation of AT under physiological conditions, as well as their alterations in particular pathological settings, are closely intertwined.

### 3.1. BAT

BAT has a characteristic yellow-brown color due to a multilocular arrangement of lipid droplets, high content of mitochondria, and high vascularity, but what makes it deeply different from white adipocytes is that the oxidation of free fatty acids is not necessarily related to ATP production. In BAT, energy is also used to produce heat. This process is called non-shivering thermogenesis and results mainly from the activity of mitochondrial uncoupling protein 1 (UCP1), a protein located within the mitochondrial matrix, which generates heat by uncoupling electron transport from ATP production, leading to a controlled exothermic resolution of the electrochemical gradient. BAT can both store lipids and dissipate energy as heat, triggering ATP-consuming futile cycles (Proton leak via ADP/ATP carrier, calcium, creatine-phosphocreatine, lipid cycling) [27,28]. In general, the regulation of non-shivering thermogenesis can occur through several mechanisms, some of which are still to be clarified. The main and most described mechanism involves adrenergic stimulation and an abundance of long-chain fatty acids that trigger a competitive reaction with purine nucleotides that modulate UCP1 activity [29]. Furthermore, β-adrenergic stimulation and heat stress produce high levels of reactive oxygen species (ROS) able to support thermogenesis [30]. While WAT localization shows clear sexual dimorphism, BAT deposits are similar in both genders [31]. The volume and activity of BAT are greatest in the supraclavicular, axillary, cervical, paraspinal, mediastinal, and abdominal anatomical districts. More than 67% of activated BAT, after 5 h of cold exposure in healthy young men, was located in the upper back of the body. These BAT deposits represent 4.3% of total fat mass and 1.5% of total body mass [32]. Retrospective studies show that, although of similar localization, BAT has a higher mass and activity in women [33,34]. The reduced muscle mass in infants and children makes BAT essential for its thermogenic activity. Clinical evidence reveals an inverse correlation between BAT and age with higher levels confirmed in young adults, gradually decreasing with aging [35,36]. A similar inverse relationship was also observed between BAT and body mass index (BMI) suggesting a correlation with overweight and obesity [36,37]. In general, the amount of BAT in adults is quite low, decreases with aging, and is inversely proportional to BMI [33,38]. Considering its functional skills, the modulation of BAT and its molecular pathways could represent interesting targets in pathological conditions affecting AT, such as metabolic syndrome and obesity.

### 3.2. Browning

WAT and BAT undergo dynamic adaptive changes in response to exogenous stimuli or deficiencies. Exposure to cold or thermoneutrality, as well as starvation or overfeeding, can first trigger alterations in protein expression and then lead to qualitative and quantitative tissue remodeling and the modification of functional properties [9,39]. WAT partially modifies its structure towards a beige phenotype, with intermediate characteristics compared to BAT and WAT, partially changing its function and capabilities. The process of acquiring a thermogenic phenotype, closer to BAT, is called browning [40,41]. BAT and BeAT share many functional features mainly related to the energy dissipation capacity (Figure 1). Beige adipocytes exhibit lipid deposits and a mitochondrial density midway between WAT and BAT. The beige phenotype is defined as brown-like due to the presence of UCP1 (missing in WAT and widely expressed in BAT) and the consequent energy dissipation capabilities [42]. However, from a localization point of view, they differ substantially, because BeAT is mainly localized in the WAT deposits, sharing a common lineage of progenitor cells [43,44]. While brown adipocytes derive from progenitor cells common to myocytes, beige adipocytes develop from a lineage shared with white adipocytes [43,44]. Specifically, experimental evidence supports that white and beige adipocytes derive from preadipocytes, in turn developing from endothelial and perivascular cells, demonstrating a close correlation between adipogenesis and angiogenesis during the expansion of AT [45]. Other experimental works suggest different potential precursors located in different districts, underlining the complexity behind understanding the different types of adipogenic lineages [46]. The browning process may favor the de novo production of beige adipocytes starting from specific precursors localized mainly in WAT deposits [46,47,48] or referring to a transdifferentiation mechanism charged to white adipocytes themselves [49]. Furthermore, WAT and BeAT exhibit an interesting plasticity. They can modify their own morphology and function depending on exogenous or endogenous stimuli [50]. The experimental evidence shows that beige adipocytes develop from precursors in the inguinal WAT (iWAT) following exposure and adaptation to cold, may switch their morphology and gene expression again to a white adipocyte phenotype under thermoneutrality or adaptation to heat (whitening). The same white adipocyte cells regain a beige morphology after a second exposure to cold through a process of transdifferentiation (Figure 2) [51].

## 4. Polyphenols and Sirtuins Interaction as a Tool for Browning Stimulation

As previously described, pathologies affecting AT lead to tissue remodeling accompanied by adipocyte cell hypertrophy, angiogenesis alterations, recruitment of immune cells that switch from alternately activated macrophage (M2) to classically activated macrophage (M1), and release of pro-inflammatory mediators [52]. The different quantitative distribution of AT phenotypes, their localization, and the plasticity of progenitor cells and mature adipocytes are all interconnected and are important aspects to consider for the evaluation of cellular and molecular changes affecting AT in pathological conditions [40,53]. In this complex scenario, the browning process has a double scientific interest: On the one hand, different pathological conditions such as obesity, metabolic disorders, T2D, cardiovascular pathologies, or predisposing factors such as physiological ageing, can be accompanied by an inhibition of the browning process, exacerbating or even causing further pathological imbalances [54]. On the other hand, the exogenous stimulation of browning could represent a string to our bow against several metabolic pathologies. Growing evidence has highlighted the ability of polyphenols to promote browning of adipocytes and improve metabolic homeostasis (detailed in paragraph 5). Polyphenols are naturally occurring compounds found in many fruits and vegetables. Polyphenols include different chemical classes of compounds. Among these, edible fruits from the genus Citrus mainly contain flavonoids; red grapes are rich in catechins, anthocyanins, stilbenes, and tannins; apples contain procyanidins, epicatechins, and catechins; cereals and legumes contain flavonoids and phenolic acids; cocoa is rich in catechins, procyanidins, anthocyanins, and flavonol glycosides [55]. They share many beneficial effects in a wide range of organs and tissues, exhibiting antioxidant, anti-inflammatory, and anti-senescence properties [56,57,58,59,60,61,62]. Recently, the potential activity of polyphenols in the regulation of lipid metabolism has also aroused particular interest [37,63]. Among the different pathways modulated by polyphenols and those involved in the regulation of the browning process, sirtuins represent interesting players (Figure 1). Molecular docking analysis suggests that sirtuin-activating compounds bind a three-helix bundle N-terminal domain, resulting in an activating effect due to a direct binding to the enzyme. In this regard, the direct binding of resveratrol to the SIRT1 isoform promotes protective effects by regulating antioxidant responses. Further information on the protein–protein interaction highlights the involvement of other binding factors (such as Lamin A) able to synergize with resveratrol to activate SIRT1 [64]. For this reason, the main purpose of this review is to elucidate the effect of polyphenols on the browning process through the modulation of sirtuin’s pathway.

### 4.1. Sirtuins in Adipose Tissue

Sirtuins are nicotinamide adenine dinucleotide (NAD+) class III histone deacetylases [65]. There are seven different mammalian isoforms (SIRT1-7) containing conserved catalytic and NAD^+^-binding domains and involved in the control of several cellular physiological processes, including cell survival, senescence, and metabolic homeostasis [66]. Increasing evidence supports the crucial role of sirtuins in regulating pathophysiological processes of WAT, such as lipid mobilization, inflammation, fibrosis, differentiation process, and browning. Sirtuins differ in tissue and subcellular localization. SIRT1 is mainly localized in the nucleus. In AT, it regulates inflammatory processes, cellular senescence, apoptosis, autophagy, and mitochondrial biogenesis [67]. SIRT2 is expressed in both the nucleus and cytoplasm and mainly regulates processes of adipocyte differentiation, gluconeogenesis, insulin sensitivity, and inflammation [68,69]. SIRT3 is mainly localized at the mitochondrial level where it regulates mitochondrial biogenesis and influences the processes of energy production and thermogenesis [70,71]. SIRT4 and SIRT5 are mainly localized in the mitochondria, while SIRT6 and SIRT7 have a nuclear localization (Figure 2). The expression levels of SIRT1 and SIRT7 are higher in WAT; SIRT3 and SIRT5 are mostly expressed in BAT; meanwhile, the expression of SIRT2, SIRT4, and SIRT6 are comparable in WAT and BAT [72].

### 4.2. Sirtuins and Browning

Sirtuins modulate AT remodeling through the regulation of adipogenesis [73], inflammatory processes [74], lipid mobilization [75,76], and fibrosis [77]. For example, SIRT1 suppresses adipogenesis and promotes lipolysis by modulating the PPAR-γ and the AMP-activated protein kinase (AMPK) [78,79]. SIRT1 exerts anti-inflammatory effects by modulating nuclear factor kappa-light-chain-enhancer of activated B cells (NF-kB) and fibrosis containing extracellular matrix over-production [79,80,81]. SIRT2 inhibits adipogenesis by deacetylating the FOXO1 and mediates the increase in fatty acid oxidation [82]. SIRT3 and SIRT6 reduce lipid deposition by activating the AMPK pathway [83]. SIRT4 represses fatty acid oxidation by deacetylating malonyl CoA decarboxylase [84]. SIRT5 affects lipid deposition and differentiation and SIRT7 promotes adipogenesis by inhibiting the autocatalytic activation of SIRT1 [85,86]. Sirtuins are linked to processes of preadipocyte differentiation towards mature white, brown, or beige phenotypes. Increasing experimental evidence also correlates sirtuins with the browning and thermogenesis processes [87]. Regarding the ability to modulate the browning process, SIRT1 is the most studied and characterized isoform. Recent studies also underline the involvement of other sirtuins (SIRT2, SIRT3, SIRT5-7) [88,89,90,91] with the exception of SIRT4, for which, at the moment, no studies have demonstrated or investigated its possible involvement in the browning mechanism and in thermogenesis processes.

#### 4.2.1. SIRT1

Exogenous modulation of SIRT1 promotes mitochondrial biogenesis, thus suggesting the possible involvement of the enzyme in the regulation of the browning process (Figure 2) [92,93]. SIRT1 induces white adipocyte browning in vitro and in vivo via the deacetylation of PPAR-γ, promoting energy expenditure, thermogenic activity, and insulin sensitivity. This event also involves the activation of co-factor PR domain-containing protein 16 (PRDM-16) and suppression of the nuclear receptor corepressor (NCoR) as well [92]. In addition, SIRT1 modulates PGC-1α. PGC-1α plays an important role in mitochondrial biogenesis and is particularly expressed in skeletal muscle (SKM) and AT [94]. It also plays a key role in regulating thermogenesis in BAT and the browning process in WAT [88,95]. SIRT1 interacts with PGC-1α, through the involvement of PPAR-γ, allowing the induction of genes characteristic of BAT and the inhibition of genes of WAT during the transdifferentiation process in white adipocytes [92,96]. SIRT1 induces adaptive thermogenesis on WAT via the AMPK/SIRT1/PGC-1α pathway in a mouse model of metabolic syndrome induced by a high-fat high-sucrose (HFHS) diet [97]. Activation of the AMPK/SIRT1 pathway promotes AT remodeling and thermogenesis in vivo through PPAR-γ deacetylation [98]. Experimental evidence in in vivo models highlights how the SIRT1/AMPK/PPAR-α pathway, activated through the inhibition of nicotinamide N-methyltransferase (NNMT), contributes to the positive regulation of the browning process [99]. Interesting experimental data obtained from mouse models demonstrate the direct involvement of Ca^2+^ influx through transient receptor potential vanilloid 1 (TRPV1) channels for the activation of Ca^2+^/calmodulin-activated protein kinase II (CaMKII) and AMPK and SIRT-1 phosphorylation, thus leading to the SIRT1-dependent deacetylation of PPARγ and PRDM-16 modulating energy metabolism and the browning process on WAT [100]. SIRT1 deficiency in BAT is associated with a significant reduction in mitochondrial biogenesis, UCP1, and PGC-1α and a down-regulation of fatty acid beta-oxidation genes, such as PPAR-α, resulting in reduced thermogenic activity [87]. The activation of SIRT1 and β3-adrenergic receptors stimulates lipid mobilization and browning in WAT and thermogenesis in BAT [101]. The energetic activity of WAT, mitochondrial biogenesis, oxidative processes, thermogenesis, and browning are closely related to SKM activity. SIRT1 deacetylates PGC-1α in SKM. AMPK is responsible for the phosphorylation of PGC-1α and is closely associated with the deacetylase activity of SIRT1 [102,103,104]. The activation of PGC-1α in SKM, but also in other tissues (with different results depending on the tissue), leads to the proteolytic cleavage of fibronectin type III domain-containing protein 5 (FNDC5) resulting in the release of irisin, a myokine associated with the stimulation of the browning process and increased energy expenditure in WAT [95].

#### 4.2.2. SIRT2

The SIRT2 isoform in AT is mainly involved in adipogenesis regulation. SIRT2 is overexpressed in WAT following caloric restriction and in BAT following cold exposure. The experimental evidence suggests that an overexpression of SIRT2 in WAT is responsible for the inhibition of adipogenesis under conditions of energy need. This mechanism is PPAR-γ/FOXO1-dependent [82]. This scenario could therefore highlight how, under conditions of extreme cold, SIRT2 contributes to the thermogenesis process in BAT and to the inhibition of adipogenesis under caloric restriction in WAT. Further experimental evidence demonstrates that an overexpression of SIRT2 has effects superimposable to those mediated by SIRT1 (i.e., reduction in adipogenesis in WAT) [105]. Still to be elucidated, this experimental evidence could indicate that the SIRT2 inhibition of adipogenesis and reduction in lipid deposits in WAT are accompanied by the onset of beige adipocyte phenotype and, therefore, by stimulation of the browning process (Figure 2).

#### 4.2.3. SIRT3

SIRT3 is localized within the inner mitochondrial membrane and is mainly expressed in BAT. Its localization suggests a predominant action in thermogenesis. SIRT3, through both ADP-ribosyl transferase and deacetylase activities, directly or indirectly (CREB phosphorylation) promotes the expression of PGC-1α and UCP1 in BAT [72,106]. SIRT3-KO mice showed the whitening of BAT accompanied by a shift from multilocular to unilocular morphology and increased lipid content in high-salt murine models [107]. Further studies showed that inhibition of the AMPK/SIRT3 pathway promotes the whitening process in BAT [89]. PGC-1α induces SIRT3 expression in white adipocytes and embryonic fibroblasts, highlighting a crosstalk that promotes a genetic pattern characteristic of a brown-like phenotype [108]. SIRT1 and SIRT3 are required for AMPK/PGC-1α activation, as demonstrated by the results of experiments on 3T3-L1 cells [109,110]. The transdifferentiation of white adipocytes into beige adipocytes is promoted by SIRT3, which acts on UCP1 through the involvement of PGC1-α, PRDM-16, and NRF1 in both 3T3-L1 cells and iWAT of mice exposed to cold. The browning process was abolished in SIRT3-KO, suggesting that the beige remodeling of WAT is regulated by SIRT3 (Figure 2) [111].

#### 4.2.4. SIRT5

SIRT5 is mainly expressed in BAT and influences its function by regulating the mitochondrial respiration process. The activity of SIRT5 in WAT and the influence on the browning process provide conflicting results. Experimental evidence supports the hypothesis that SIRT5 acts positively on the regulation of brown adipocyte differentiation in vitro and the browning process in vivo. SIRT5 deficiency in mice results in decreased thermogenic capacity in subcutaneous WAT (sWAT). In iWAT from SIRT5-KO mice, the expression of thermogenic genes, including UCP1, was down-regulated [90]. The inhibition of SIRT5 in the early stages of differentiation promoted the browning process through enhanced AMPK activation in 3T3-L1 cells and primary cell cultures of murine preadipocytes [112]. Furthermore, SIRT5 inhibited the thermogenesis process in BAT by succinylation of UCP1 (Figure 2) [113].

#### 4.2.5. SIRT6

The main effects of SIRT6 on AT are to regulate the inflammatory process and lipid metabolism. SIRT6 increases the thermogenic function of BAT and browning of sWAT by recruiting the promoter of PGC-1α gene [114,115]. SIRT6 deficiency inhibits browning in WAT following cold exposure or β3-agonist treatment. SIRT6 overexpression in primary adipocytes stimulates thermogenesis by activating PGC-1α expression through phospho-ATF2 (pATF2) [115]. Low UCP1 levels associated with increased adipocyte size were observed in BAT and iWAT from SIRT6-KO mice. The mRNA levels of beige/brown adipocyte marker genes (UCP1, *PPARGC-1α*, PRDM-16, CIDEA, and Elovl3) were significantly down-regulated in BAT and iWAT from SIRT6-KO mice (Figure 2) [91].

#### 4.2.6. SIRT7

The experimental evidence suggests that SIRT7 deficiency promotes browning in the iWAT of SIRT7-KO mice, as pointed out by the increased expression of Elovl3, type II iodothyronine deiodinase (DIO2), PGC-1α, and UCP1. Administration of norepinephrine (activator of adrenergic receptors) to SIRT7-KO mice clearly showed that SIRT7 deficiency increases browning in the iWAT (Figure 2) [116].

## 5. Dietary Polyphenols and the Browning Process: Evidence from Pre-Clinical Studies

This section is divided into two sub-sections: the first one (Section 5.1) provides evidence about the effect of polyphenols in inducing the browning process, considering the most studied biomarkers of brown-phenotype adipose tissue, such as UCP1 and PGC-1α. They regulate thermogenesis, enhance energy expenditure, and promote oxygen consumption, which represent the main functions of BAT. The second sub-section (Section 5.2) examines how polyphenols can induce the upregulation of UCP1 and PGC-1α, focusing the attention on SIRT1. Pre-clinical studies support the hypothesis that polyphenols act mainly through the activation of SIRT1 in inducing browning process. The potential role of SIRT1 as the epigenetic regulator of adipose function associated with the browning process is a burning and challenging field of pharmacological research, which could pave the way for innovative strategies for the prevention/treatment of adipose tissue dysfunction.

### 5.1. Polyphenols and Browning

Many studies have demonstrated the biological effects of dietary polyphenols in cultured adipocytes (Table 1). Micromolar concentrations of resveratrol prevented preadipocyte differentiation, promoted mature adipocyte lipolysis [117], and decreased lipid accumulation in murine (3T3-L1) and human (SGBS) adipocytes [118]. Both isolated polyphenols (e.g., apigenin, hesperidin, myricetin, quercetin, resveratrol) and polyphenol extracts inhibited adipogenesis through modulation of PPAR-γ expression [117,119,120,121,122,123], but other targets may be involved in the pharmacology of natural polyphenols. For instance, naringenin and naringin (20–100 µM) decreased triglyceride and lipid accumulation in 3T3-L1 cells by phosphorylating AMPK and acetyl-CoA carboxylase (ACC) [124], while resveratrol (50–100 µM) reduced lipid droplets in cultured adipocytes by decreasing AMPK and ACC levels [118]. In recent years, the potential modulation of the browning process by natural polyphenols has emerged. A novel mix of micronutrients and polyphenols (e.g., polydatin, pterostilbene, and honokiol) reduced adipogenesis and promoted the browning process by increasing AMPK, CIDEA, and UCP1 expression in cultured 3T3-L1 cells [125]. However, these effects cannot be ascribed exclusively to the polyphenols, as many micronutrients (zinc, selenium, and chromium) were present in the mixture. To confirm the role of polyphenols in the process of thermogenesis, other studies should be mentioned. For instance, resveratrol (10–50 µM) reduced lipid accumulation and induced brown fat-like phenotype in 3T3-L1 adipocytes by enhancing the expression of PPAR-γ, PGC-1α [126], and UCP1 [123,126]. Naringenin (8 µM) up-regulated the expression of genes and proteins involved in the process of thermogenesis (e.g., PGC-1α, PGC-1β, p-AMPK, and UCP1) in human adipose-derived stem cells (hADSC) and sWAT from overweight female donors [127]. The resveratrol derivative oxyresveratrol (100 µM), found in *Morus alba* L., reduced lipid accumulation and increased mitochondrial mass in 3T3-L1 cells by decreasing PPAR-γ expression and up-regulating UCP1 [128], while 6-gingerol (20 µg/mL), a ginger polyphenol, promoted mitochondrial respiration and energy metabolism in 3T3-L1 adipocytes via a PPAR-γ/AMPK/PGC-1α/UCP1-dependent pathway [121]. It is noteworthy that the up-regulation of UCP1 induced by the polyphenol quercetin (10 µM) in 3T3-L1 cells was partially inhibited by a PPAR-γ antagonist [129]. Moreover, in differentiated iWAT stromal vascular cells, resveratrol (10–40 µM) prevented lipid accumulation and increased oxygen consumption by enhancing the expression of PRDM-16, PGC-1α, CIDEA, p-AMPKα1, and UCP1, but these effects were not observed in AMPKα1-lacking cells [130]. Taken together, these results indicate that the biological properties of polyphenols in AT could result from the modulation of multiple targets (i.e., AMPK, PPAR-γ) ultimately leading to increased UCP1 levels and activation of the browning process. Overall, the effects on adipogenesis do not seem to depend on the stage of differentiation in vitro, contrary to what was suggested in a recent study [122], since there are no relevant differences in the pharmacology of polyphenols when added at the initial stage of differentiation or when cells are already differentiated (transdifferentiation; Table 1).

The promising effects of natural polyphenols in the AT were also confirmed by in vivo studies (Table 2). This evidence indicates that dietary polyphenols are potential candidates for the prevention and treatment of metabolic syndrome and related disorders. In this regard, daily treatment with bergamot juice (12% v/v for 3 weeks) or mandarin juice (24% v/v for 3 weeks), particularly rich in polyphenols, reduced body weight gain and WAT weight in a rat model of overweight/obesity [131,132]. Oral treatment with resveratrol for 10–12 weeks suppressed diet-induced obesity in mice [117,133,134], partially through activation of fatty acid oxidation [134] and reduction in fatty acid biosynthesis [133]. Furthermore, a 12-week treatment with lemon polyphenols from lemon peel (0.5% w/w) significantly reduced body weight, blood glucose levels, serum triglycerides, and insulin resistance in mice fed with a high-fat (HF) diet by increasing the mRNA levels of acyl-CoA oxidase (ACO) and fatty acid synthase (FAS) in epidydimal WAT (eWAT) [135]. The potential role of the browning process in the beneficial effects of polyphenols against diet-induced metabolic syndrome has been largely investigated. Daily treatment with resveratrol (0.4% w/w for 6 weeks) reduced WAT weight and induced WAT browning by the up-regulation of PRDM-16, CIDEA, PPAR-γ, and UCP1 in db/db mice [136]. Similarly, resveratrol (0.1% w/w for 4 weeks) [130] and oxyresveratrol (7.5–15 mg/kg for 8 weeks) [128] prevented body weight gain and promoted the browning process by increasing oxygen consumption and activating the PRDM-16/PGC-1α/UCP1 pathway in the iWAT of diet-induced overweight mice [128,130]. However, the low oral bioavailability represents a strong limitation for the use of resveratrol in clinical practice. Transdermal delivery of trans-resveratrol to adipose stromal stem cells for 4 weeks reduced body weight and increased UCP1 mRNA expression in the WAT of rats fed with a HF diet, suggesting that this delivery technology may represent an efficient and innovative approach to induce browning of sWAT and prevent obesity [137]. The potential effects of polyphenols on the browning process are not limited to resveratrol and its derivatives. Dietary supplementation with quercetin (0.05% w/w for 9 weeks) significantly induced the brown fat-like phenotype by enhancing the expression of PRDM-16, CIDEA, PPAR-γ, PGC-1α, and UCP1 in the WAT of mice fed with a HF diet [129]. In diet-induced overweight rats, both the polyphenol oleuropein (1–4 mg/kg for 4 weeks) and the isoflavone daidzein (50 mg/kg for 2 weeks) reduced body weight and enhanced UCP1 expression in the BAT [138,139]. Grape polyphenols (1.0% w/w for 23 weeks) increased energy expenditure, as measured by direct calorimetry and reduced adiposity in overweight mice. It is noteworthy that an inverse relationship was observed between body weight gain and UCP1 expression in the BAT of mice receiving grape polyphenols [140]. Oral administration of theaflavins (10 mg/kg for 2–20 h), the most abundant polyphenols in black tea [141], or (−)-epigallocatechin gallate (1.0% w/w for 4 weeks) [142] led to a marked increase in PGC-1α and UCP1 mRNA levels in BAT. This mechanism may be responsible for the increased in energy expenditure recorded in fasted animals placed in an open-circuit metabolic chamber [141]. Furthermore, a 4-week treatment with the flavan-3-ol fraction of cocoa powder (0.2% w/w) significantly increased the UCP1 protein levels in the BAT of rats fed with a HF diet, thereby reducing eWAT weight in the experimental model of metabolic syndrome [143]. Daily treatment with three different flavonoid compounds (elderberry, blackcurrant, and aronia berry extract powders, 1.0% w/w for 16 weeks) significantly reduced body weight gain and fat mass in mice fed with HF diet. Of note, after the consumption of dietary flavonoids, the portal plasma concentration of the intestinal microbial metabolite 4-hydroxyphenylacetic acid (4-HPAA) increased and negatively correlated with body fat percentage. The potential effects of the flavonoid metabolite 4-HPAA on the browning process were therefore investigated. 4-HPAA (350 µg for 6 weeks), delivered from implanted subcutaneously delivered pellets, significantly up-regulated PGC-1α and UCP1 in the BAT of mice fed with a HF diet, confirming the potential role of the flavonoid metabolite in the regulation of thermogenesis [144]. Consumption of apple polyphenols (5.0% w/w for 4 weeks) decreased AT mass, induced beige adipocyte development, and promoted thermogenic adaptations in the inguinal, but not visceral, WAT of mice fed with a HF diet, through increased gene expression of UCP1, as well as the protein content of UCP1 and mitochondrial oxidative phosphorylation system (OXPHOS) enzymes. This effect may result from the up-regulation of PGC-1α and fibroblast growth factor 21 (FGF21) in iWAT, which are two up-stream regulators of the browning process [145]. Accordingly, downstream signaling cascades were also activated (i.e., phosphorylated/active form of fibroblast growth factor receptor substrate 2α (FRS2α); extracellular signal-regulated kinase 1/2 (Erk1/2); p38 mitogen-activated protein kinase (MAPK)). Of note, these effects were not observed in cultured adipocytes, as apple polyphenols did not induce thermogenic adaptations (i.e., an increased gene expression of PGC-1α, FGF21, or UCP1) during differentiation from adipose-derived stem cells into mature adipocyte. This evidence suggests that the effects on the browning process could derive from the modulation of additional pathways in vivo [145] and supports the hypothesis of a possible role of the regulator irisin, released by SKM, in the pharmacology of polyphenols. In this regard, a procyanidin extract of cocoa liquor (total polyphenols: 69.8% w/w) prevented hyperglycemia and obesity in mice fed with a HF diet by phosphorylation of AMPK and enhanced expression of PGC-1α, with subsequent up-regulation of UCP1 in BAT, uncoupling protein 2 (UCP2) in WAT and uncoupling protein 3 (UCP3) in SKM [146]. An increased expression of both UCP3 in SKM and UCP1 in BAT was also demonstrated at the end of a 4-week treatment with flavan-3-ols from cocoa powder (0.2% w/w) [143] and after acute treatment with theaflavins (10 mg/kg) [141]. Worthy of note, the daily intake of polyphenol-rich bergamot juice (24% v/v for 3 weeks) reduced PPAR-γ expression and up-regulated PGC-1α and UCP1 genes in the WAT of rats fed with a HF diet, probably by increasing plasma levels of the myokine irisin [131]. However, in another study, resveratrol (1–25 µM) did not up-regulate FNDC5, the precursor of irisin, in C2C12 mouse myoblasts, indicating that further studies are needed to better elucidate the potential role of irisin in the pharmacological effects of dietary polyphenols [147]. Moreover, to the best of our knowledge, there are no studies on UCP1 knock-out animals, thus not allowing the confirmation that the promising anti-obesity effects of dietary polyphenols derive exclusively from the modulation of the browning process in vivo.

### 5.2. SIRT1-Mediated Browning Effects of Polyphenols 

The possible therapeutic effects of dietary-derived polyphenols in counteracting and preventing AT dysfunction are gaining great interest. As mentioned above, polyphenols promote the differentiation from white to beige/brown AT by activating different and heterogeneous intracellular pathways. However, the mechanism of action responsible for the regulation of the metabolic profile of adipose cells has not been elucidated yet. In the last few years, epigenetic modulation is emerging as fundamental process in the control of several biological pathways. Among epigenetic factors, sirtuins seem to play a pivotal role. As previously reported, SIRT1 mediates thermogenesis and the brown-like remodeling of WAT in vitro (using 3T3-L1 adipocytes) and in vivo using SirBACO mice and Dbc1^−/−^ and Sirt1^−/−^ mice, highlighting the role of SIRT1 in deacetylating PPAR-γ on Lys293 and Lys268 residues. Indeed, SIRT1 gain-of-function, by deacetylating PPAR-γ, promoted the recruitment of PRDM-16, which controls the development of brown adipocytes in BAT [92]. Due to this intriguing connection between SIRT1 and browning process, the activation of this epigenetic factor could represent a very promising strategy to prevent and reduce the onset of AT dysfunctions. Many natural polyphenols are SIRT1 activators and have been widely investigated for their ability to modulate adipose cell metabolism [148] (Table 3 summarizes the studies supporting this hypothesis). Resveratrol activates SIRT1 by mediating a plethora of beneficial effects including antioxidant, anti-inflammatory, anti-carcinogenic, and anti-adipogenic effects, resulting in increased mitochondrial respiration and enhanced energy expenditure [58,149,150]. Indeed, resveratrol promoted aerobic capacity (which consists of higher oxygen consumption and longer running time) in mice fed with a HF diet and KKay mice treated with resveratrol added to food, supplying a dose of 200 or 400 mg/kg/day. The mechanism of action involves SIRT1-mediated increased activity of PGC-1α, leading to enhanced mitochondrial biogenesis and oxidative phosphorylation, as also demonstrated in C2C12 cells [151]. Another study demonstrated that mice fed with a diet containing 22.5% of fat and 20.0% of sucrose, but ensuring resveratrol 30 mg/kg/day, showed increased expression of SIRT1, PGC-1α, UCP1, mitochondrial transcription factor A (TFAM)—a marker of mitochondriogenesis—and COX2—a key factor for the oxidative phosphorylation pathway in BAT [152]. The axis polyphenols–SIRT1–browning was also evident in a recent study, where resveratrol decreased the lipid deposit in 3T3-L1 adipocytes by activating SIRT1. Moreover, resveratrol induced a browning-like phenotype in WAT in in vivo models (KKAy and diet-induced obese C57BL/6J mice models). SIRT1 knockdown completely abolished this effect, thus reducing the resveratrol-induced activity of adipose triglyceride lipase (ATGL) and PRDM-16 proteins, providing evidence for the involvement of SIRT1 in the browning process. Of note, resveratrol regulated lipid and glucose catabolism in a SIRT1-dependent manner [79]. More recently, a further study provided another piece of evidence in demonstrating the close connection between SIRT1 activation and the browning process in mice fed with a HF diet, supplying resveratrol 400 mg/kg. The most interesting result concerns the effect of resveratrol in improving lipid and glycemic status simultaneously with SIRT1, PGC-1α and PRDM-16 increased expression. This study also reported the up-regulation of FNDC5 in mice adipose tissues, paving the way for another interesting crosstalk involving SIRT1 activity. Indeed, SIRT1 knockdown resulted in a significant down-regulation of FNDC5 [153]. In support of these results, the effect of SIRT1-deficiency in the adipose tissues of HF diet-induced obese mice was explored. SIRT1-deficient mice showed a marked reduction in heat production and oxygen consumption; furthermore, histochemical analysis with hematoxylin/eosin staining revealed the increased dimension of lipid droplets in adipose tissues of SIRT1 deficient mice compared to the controls. Furthermore, a significant reduction in UCP1 expression was reported [154]. Beside mice models, the effect of resveratrol supplementation on the browning process was also demonstrated using rhesus monkeys in a regimen of 80 and 480 mg/kg/day for two years. Resveratrol decreased adipocyte size, promoted SIRT1 up-regulation, and reduced chronic inflammation in WAT. Moreover, resveratrol promoted the expression of thermogenic and insulin markers. This effect was also reproduced in vitro in 3T3-L1 cells treated with serum derived from resveratrol-fed rhesus monkeys [155]. Other polyphenols have been reported to promote browning-like differentiation of WAT in a SIRT1-dependent manner. Another study, in fact, demonstrated that 3T3-L1 cells treated with genistein, a naturally occurring polyphenol (belonging to an isoflavone chemical structure) found in soy products, enhanced oxygen consumption and increased UCP1 expression by promoting SIRT1 activity [156]. Epicatechin, a polyphenol contained in green and white tea, and cocoa, showed positive effects in mice fed with a HF diet. In particular, a 2-week treatment with epicatechin 1 mg/kg body weight showed reduced weight gain, up-regulation of UCP1, PGC-1α, and SIRT1 in WAT, and reduced plasma triglyceride levels in mice [157]. AT dysfunction is also characterized by a massive production of pro-inflammatory cytokines resulting in chronic inflammation that could contribute to the onset of cardiovascular pathologies. The administration of quercetin, a polyphenol ubiquitously contained in vegetables and fruits, has been shown to reduce AT macrophage infiltration in mice fed with a HF diet. After 12-week, treatment with quercetin 0.1% reduced body weight gain, insulin resistance, and glucose plasma levels. Furthermore, serum inflammatory biomarkers (TNF-α and IL-6) were reduced. A specific analysis of AT was also performed: SIRT1, UCP1, and PCG-1α activity was increased in AT from mice treated with quercetin [158]. Quercetin (240 mg/kg/day) was also administered in association with resveratrol (120 mg/kg/day) to rats fed with a HF diet for 8 weeks, resulting in reduced body weight gain. Furthermore, the combination of the two polyphenols promoted a significant decrease in plasma cholesterol and triglyceride levels, thus limiting cardiovascular risk factors. Inflammation was also reduced: a lower level of circulating monocyte chemoattractant protein-1 (MCP-1), IL-6 and TNF-α was reported in resveratrol + quercetin-treated group. These effects were mediated by the increase in SIRT1/AMPK activity in AT [159].

## 6. Polyphenols and Human Obesity: The Still Unconsidered Role of Sirtuins

To our knowledge, there are no clinical studies that provide a mechanistic investigation of the effects of dietary polyphenols in the regulation of the browning process. The BEACON BEAMS study, an ongoing open label randomized clinical trial (RCT) [160], was designed to evaluate the efficacy of curcumin supplementation (2 g/day for 12 weeks) in the activation of BAT in obese patients. However, the potential role of SIRT1 in the pharmacology of curcumin in human adipose tissue has not been considered. In another study, daily treatment with resveratrol (3 g/day for 12 weeks) enhanced energy expenditure by increasing SIRT1 expression and p-AMPK/AMPK ratio in the SKM of overweight patients with type 2 diabetes [161]; however, the possible involvement of SIRT1 in the effects of resveratrol in the adipose tissue has not been investigated. Finally, the POLYGIR study [162] is an ongoing, double-blind, placebo-controlled RCT aimed at assessing the effect of red grape polyphenols supplementation (dosage unknown) on metabolic parameters in obese insulin-resistant subjects. Worthy of note, a secondary outcome of the POLYGIR study is the evaluation of SIRT1 gene expression in SKM and WAT of patients receiving red grape polyphenols or placebo for 8 weeks, confirming the growing scientific interest towards the potential role of sirtuins in the anti-obesity effects of polyphenols. However, the results of the POLYGIR study have not yet been published, and conclusive evidence on the role of SIRT1 in the effects of polyphenols in human adiposity is still lacking. In order to show what is known and what is missing in this promising research field and to partially overcome the main limitation of our review (i.e., the lack of clinical evidence about the role of SIRT1 in the beneficial effects of polyphenols), we will provide a brief overview of the potential anti-obesity properties of edible plants containing polyphenols and their extracts, even if without direct correlations with SIRT1 expression/activity. Observational studies showed a positive association between high dietary polyphenol intake and prevention of BMI increase in women [163] or hip circumference/waist-to-hip ratio in overweight and obese women [164]. The effects of dietary polyphenols in overweight/obese patients were also demonstrated in clinical trials. A 3-month treatment with green tea extract (379 mg/day for 12 weeks), which contains high amount of epigallocatechin-3-gallate, reduced BMI in obese patients [165]. To rule out the potential role of caffeine, abundant in green tea, in the promising anti-obesity effects of green tea extract, an additional trial was performed. The results showed that decaffeinated green tea extract (856.8 mg/day for 12 weeks) reduced BMI and body weight in women with central obesity [166], suggesting the leading role of polyphenols in the beneficial properties of green tea. Similarly, daily consumption of grapefruit, rich in the polyphenols naringenin and hesperidin, for 6 weeks resulted in modest weight loss but significant reduction in waist circumference in overweight adults [167]. Finally, a quercetin-rich onion peel extract (100 mg quercetin/day for 12 weeks) significantly reduced body weight and fat in overweight and obese patients [168]. However, the results of studies evaluating the anti-obesity effects of isolated polyphenols, rather than whole extracts, are quite conflicting. Curcumin supplementation (500 mg/day for 10 weeks) reduced BMI and waist circumference in overweight/obese adolescent girls [169]. A recent meta-analysis of clinical trials demonstrated that resveratrol supplementation increases lean mass and decreases body weight, waist circumference, BMI, and fat mass, particularly after treatment periods longer than 17 weeks [170]. However, many of the primary studies did not include overweight/obese patients, thus not allowing further speculation on the effects of resveratrol in the obese population. A further meta-analysis showed no positive effects of resveratrol against obesity in human studies, even in the stratified analyses for time of treatment and dosage of the polyphenol [171]. A third meta-analysis revealed that long-term resveratrol supplementation significantly reduces body weight, BMI, and waist circumference in obese patients [172]. Therefore, the evidence on the potential effects of resveratrol against overweight/obesity in human studies is inconclusive, and a standardization of the experimental conditions employed is needed to better describe the pharmacology of polyphenols.

## 7. Conclusions

The weight gain control is one of the most compelling targets in a healthy lifestyle, especially in Western countries. Modulation of AT differentiation towards WAT or BAT seems to be a crucial step in maintaining optimal BMI. This review article aimed to summarize the best evidence on a novel topic which is the study of the AT browning process, the factors mainly involved in this pathway (e.g., sirtuins), and the role of polyphenols as pharmacological/nutraceutical tools capable to activate sirtuins and to promote the browning onset. The novelty of the topic, i.e., the browning process as a tool for weight gain control, represents a strength of this review work. As a further strength, all the evidence about the browning process, from molecular targets to clinical trials, have been extensively described. 

As a limitation of this work, given the novelty of the topic, there are no results from clinical trials specifically designed to investigate the influence of polyphenols on the browning process (to date, there are two RTCs, designed to target the browning process with polyphenols but they are still ongoing). On the other hand, results on the effects of polyphenols on weight loss in humans are often inconclusive or conflicting, probably due to the extreme variability in polyphenols’ formulations. Indeed, as reported in in vitro studies, polyphenols may represent powerful tools to affect body weight, AT, and other related cardiovascular or metabolic diseases, but they are characterized by poor oral bioavailability. This is mainly due to low intestinal absorption, rapid metabolism, and urinary excretion. Improving the pharmacokinetics of polyphenols is a compelling challenge nowadays, and several efforts have been made to overcome this limitation. Micronized powders are potential alternatives to optimize the stability of polyphenol solutions and improve intestinal absorption [173]. Co-administration of polyphenols with substrates of metabolizing enzymes (e.g., piperine) [174] or microencapsulation with maltodextrins/cyclodextrins [175] represent further strategies to improve the oral bioavailability of natural products. In this regard, formulations of polyphenols microencapsulated with maltodextrins are commercially available, but their use in clinical practice is limited to the antioxidant, lipid-lowering, cardio- and vaso-protective properties. Recently, it has been proposed that lipid-, protein-, or polymeric-based nanoformulations can significantly improve the pharmacokinetics of polyphenols [176], although there are still some concerns regarding their toxicity profile. This aspect will require further specific efforts to obtain technological strategies able to improve the formulation of polyphenols and ensure the standardization of the content in polyphenol-based products used in vivo and in clinical trials.

In conclusion, we can hypothesize that natural polyphenols could be effective both for the prevention and for the treatment of overweight/obesity. Indeed, they prevent the differentiation of cultured pre-adipocytes into mature adipocytes, inhibit the transdifferentiation process of mature white adipocytes in vitro and reduce body weight gain in animal models of overweight/obesity. Moreover, the regular intake of polyphenols prevents the increase in BMI in healthy subjects and promotes weight loss in overweight/obese patients. Future studies are needed to confirm the “double effect” (preventive on the one hand and therapeutic on the other) of natural polyphenols.

## Figures and Tables

**Figure 1 ijms-24-09276-f001:**
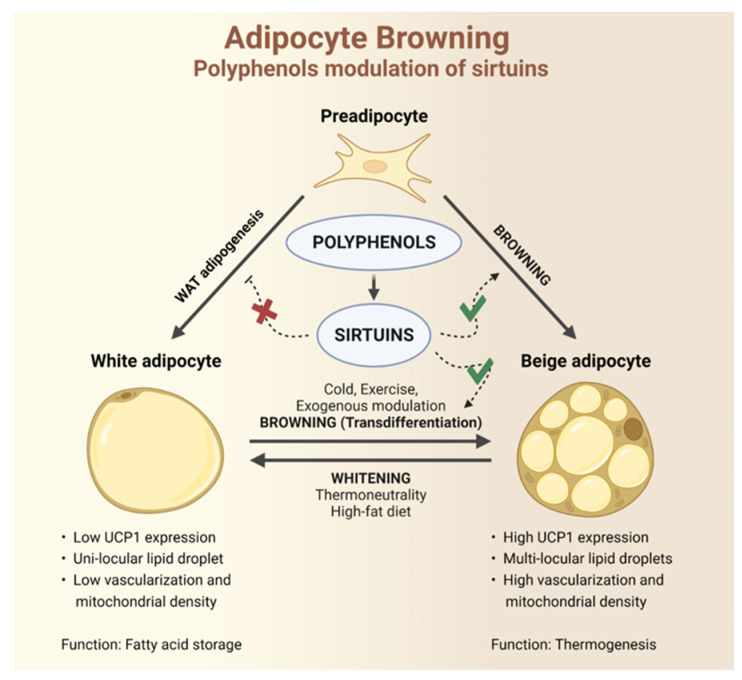
Browning process in WAT, phenotypic and functional differences between white and beige adipocytes. Polyphenols, partially through the modulation of sirtuins, can promote the browning process both through the recruitment of preadipocyte precursors and by inducing the transdifferentiation process of mature white adipocytes, while inhibiting adipogenesis towards a white phenotype. Legend: Red cross: inhibition; green check: activation.

**Figure 2 ijms-24-09276-f002:**
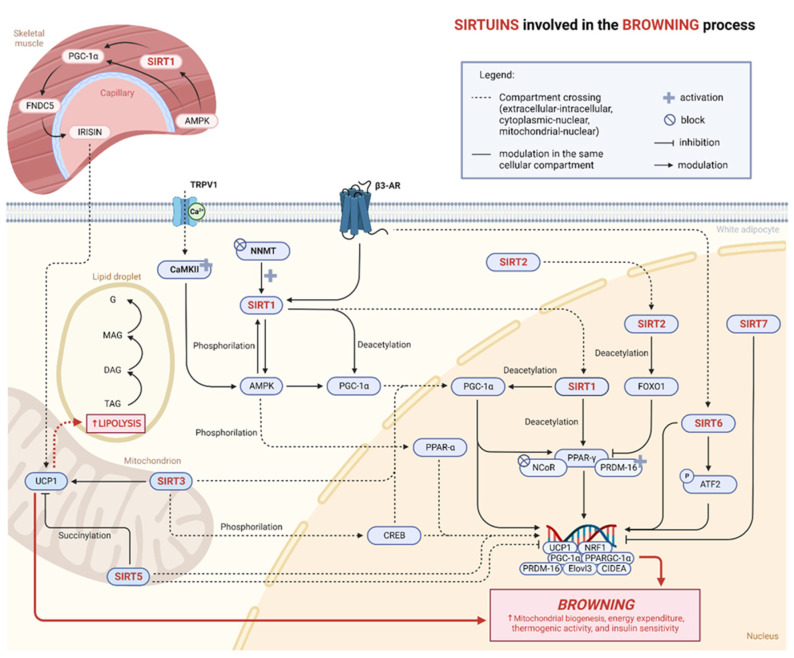
Involvement of sirtuins in the browning process. Graphical representation of the different cellular compartments of a white adipocyte and connection with skeletal muscle. Abbreviations: SIRT1–7, Sirtuin 1–7; AMPK, 5’ AMP-activated protein kinase; PGC-1α, Peroxisome proliferator-activated receptor gamma coactivator 1-alpha; PPAR-γ, Peroxisome proliferator-activated receptor gamma; PPAR-α, Peroxisome proliferator-activated receptor alpha; UCP1, Uncoupling protein 1; FOXO1, Forkhead box protein O1; ATF2, Activating transcription factor 2; NCoR, Nuclear receptor corepressor; PRDM-16, PR/SET domain 16; *PPARGC-1α*, PGC-1α encoded gene; NRF1, Nuclear respiratory factor 1; CIDEA, Cell-death-inducing DFFA-like effector a; Elovl3, ELOVL fatty acid elongase 3; CREB, cAMP response element-binding protein; NNMT, Nicotinamide N-methyltransferase; β3-AR, β-3 adrenergic receptor; CaMKII, Ca^2+^/calmodulin-dependent protein kinase II; TRPV1, Transient receptor potential cation channel subfamily V member 1; FNDC5, Fibronectin type III domain containing 5; TAG, Triacylglycerol; DAG, Diacylglycerol; MAG, Monoacylglycerol; G, Glycerol.

**Table 1 ijms-24-09276-t001:** Main findings of the in vitro studies focused on the effects of polyphenols in browning process. Abbreviations: 3T3-L1, murine preadipocyte (fibroblasts) cell line; ACC, acetyl-CoA carboxylase; C3H10T1/2, murine preadipocyte (fibroblasts) cell line; FAE, ferulic acid equivalent; FAS, fatty acid synthase; hADSC, human adipose-derived stem cells; PPAR-γ, peroxisome proliferator-activated receptor γ; pWAT, subcutaneous abdominal adipose tissue; SGBS, human preadipocyte cell line; SVCs, stromal vascular cells; UCP1, uncoupling protein 1.

First Author, Year, Reference	Cell Line	Compound, Concentration	Time	Potential Mechanism	Main Effects
Aranaz, 2019, [122]	3T3-L1	Phenolic compounds ^1,^ 10–100 µM	8 days *^,^**	↓ PPAR-γ ^a^ at the initial stage of differentiation	↓ lipid accumulation depending on the stage of differentiation
Chang, 2016, [117,129]	3T3-L1	Resveratrol, 0.03–10 µM	8 days *; 24 h **	↓ PPAR-γ ^b^	↓ preadipocyte differentiation ↑ mature adipocyte lipolysis
Choi, 2018, [129]	3T3-L1	Quercetin, 10 µM	6 days *	↑ UCP1 ^b^	-
Choi, 2018, [128]	3T3-L1, C3H10T1/2	Oxyresveratrol, 100 µM	6 days *; 24 h **	↓ PPAR-γ ^a^ ↑ UCP1 ^a^	↓ lipid accumulation and adipocyte differentiation ↑ mitochondrial mass
Dayarathne, 2021, [124]	3T3-L1	Naringenin and naringin, 20–100 µM	8 days *	↑ p-AMPK ^b^, p-ACC ^b^	↓ lipid and triglyceride accumulation
Deng, 2022, [119]	3T3-L1	Polyphenol extract, 5–25 μg FAE/ml	48 h *	↓ PPAR-γ ^b^, FAS ^a,b^	↓ lipid and triglyceride accumulation ↓ oxidative stress
Hong, 2021, [120]	3T3-L1	Quercetin, 5–20 µM	24 h **	↓ PPAR-γ ^a^	↓ fat accumulation
Li, 2016, [118]	3T3-L1, SGBS	Resveratrol, 50–100 µM	4 **–8 * days (3T3-L1), 6 **–12 * days (SGBS)	↓ AMPK ^b^, ACC ^b^	↓ lipid accumulation
Liu, 2020, [126]	3T3-L1	Resveratrol, 10–40 µM	6–8 days *	↑ PGC-1α ^b^, PPAR-γ ^b^, UCP1^b^	↓ lipid accumulation
Pacifici, 2023, [125]	3T3-L1	Mix of polyphenols ^2^, N/A	10 days *	↑ CIDEA ^a.b^, p-AMPK ^b,^ UCP1 ^a,b^	↓ lipid accumulation ↓ inflammation ↑ fatty acid oxidation
Rayalam, 2008, [123]	3T3-L1	Resveratrol, 12.5–50 µM	6 days *	↓ PPAR-γ ^a^, FAS ^a^ ↑ UCP1 ^a^	↓ lipid accumulation
Rebello, 2019, [127]	hADSC and pWAT from overweight patients	Naringenin, 8 µM	7–14 days *	↑ PGC-1α ^a^, PGC-1β ^a^, p-AMPK ^b^, UCP1 ^a,b^	-
Wang, 2015, [130]	Differentiated iWAT SVCs	Resveratrol, 10–40 µM	7 days *	↑ PRDM16 ^a,b^, PGC-1α ^a^, CIDEA ^a^, p-AMPK ^b^, UCP1 ^a,b^	↓ lipid accumulation ↑ O_2_ consumption
Wang, 2019, [121]	3T3-L1	6-Gingerol, 20 μg/ml	Until differentiation *	↑ PGC-1α ^a,b^, PRDM-16 ^a,b^, p-AMPK ^b^, UCP1 ^a,b^ ↓ PPAR-γ ^a^	↓ lipid content of mature adipocytes ↑ mitochondrial respiration and energy metabolism

Legend: * Compound added at the initial stage of differentiation. ** Compound added when cells were already differentiated (transdifferentiation). ^a^ Gene expression. ^b^ Protein expression. ^1^ Hesperidin, naringin, myricetin, kaempferol, quercetin, apigenin, luteolin, resveratrol, curcumin. ^2^ Polydatin, pterostilbene, honokiol. ↑: increased; ↓: reduced.

**Table 2 ijms-24-09276-t002:** Main findings of the in vivo studies focused on the effects of polyphenols in browning process. Abbreviations: 4-HPAA, 4-hydroxyphenylacetic acid (polyphenol metabolite); ACO, acyl-CoA oxidase; BAT, brown adipose tissue; EGCG, (−)-epigallocatechin-3-gallate; Erk1/2, extracellular signal-regulated kinase 1/2; eWAT, epidydimal white adipose tissue; FAS, fatty acid synthase; FGF21, fibroblast growth factor 21; FRS2α, phosphorylated/active form of fibroblast growth factor receptor substrate 2α; i.p., intraperitoneally; iWAT, inguinal white adipose tissue; OXPHOS, mitochondrial oxidative phosphorylation system; p38 MAPK, mitogen-activated protein kinase; pWAT, perirenal white adipose tissue; SKM, skeletal muscle; sWAT, subcutaneous white adipose tissue; WAT, white adipose tissue.

First Author, Year, Reference	Animal Model	Number (Control/ Treated)	Compound, Dosage, Route of Administration	Weeks of Treatment	Potential Mechanism	Main Effects
Abbasi, 2022, [137]	C57BL/6J mice ^#^	5/5	Peptide-resveratrol conjugate, N/A, transdermal	4	↑ UCP1 ^a^ in WAT	↓ body weight
Chang, 2016, [117]	C57BL/6C mice ^#^	10–11/ 10–11	Resveratrol, 1–30 mg/kg, orally	10	-	↓ diet-induced body weight ↓ total WAT weight ^1^
Cho, 2012, [134]	C57BL/6J mice ^#^	10/10	Resveratrol, 0.005–0.02% w/w, orally	10	↓ FAS activity in eWAT ↑ β-oxidation in eWAT	↓ body weight ↓ total WAT weight ^2^ ↓ epididymal adipocyte size
Choi, 2018, [129]	C57BL/6J mice ^#^	6/6	Quercetin, 0.05% w/w, orally	9	↑ PRDM16 ^a^, CIDEA ^a^, PPAR-γ ^b^, PGC-1α ^b^, UCP1 ^a,b^ in WAT ↑ UCP1 in BAT ^b^	↓ WAT weight and adipocyte size
Choi, 2018, [128]	C57BL/6N mice ^#^	6/7	Oxyresveratrol, 7.5–15 mg/kg, i.p.	8	↑ PRDM16 ^a,b^, PGC-1α ^a,b^, UCP-1 ^a,b^ in iWAT	↓ body weight ↓ eWAT weight ↑ energy expenditure
Crespillo, 2011, [138]	Wistar rats ^#^	16/16	Daidzein, 50 mg/kg, i.p.	2	↑ UCP1 ^b^ in BAT	↓ body weight ↓ fat depots in the liver
De Leo, 2020, [132]	Wistar rats ^#^	6/6	Bergamot juice, 12% v/v, orally	3	-	↓ body weight ↓ fat depots in the liver
Fukuchi, 2008, [135]	C57BL6/J mice ^#^	6/6	Lemon polyphenols, 0.5% w/w, orally	12	↑ FAS ^a^ and ACO ^a^ in eWAT	↓ body weight ↓ insulin resistance ↓ total WAT weight ^3^
Hui, 2020, [136]	*db/db* mice	8/8	Resveratrol, 0.4% w/w, orally	6	↑ PRDM16 ^a^, CIDEA ^a^, UCP1 ^a,b^ in iWAT/BAT ↑ PPAR-γ ^a^ in iWAT	↓ eWAT and iWAT ↓ sizes of lipid droplets in iWAT and BAT
Kudo, 2015, [141]	ICR mice	8/8	Theaflavins, 10 mg/kg, orally	Acute treatment (2–20 h)	↑ UCP1 ^a^ in BAT ↑ p-AMPK ^b^, UCP3 ^a^ in SKM ↑ PGC-1α ^a^ in BAT/SKM	↑ energy expenditure
Mezhibovsy, 2021, [140]	C57BL/6J mice ^#^	6–8/6–8	Grape polyphenols, 1.0% w/w, orally	23	Inverse relationship between body weight and UCP1 ^a^ in BAT	↓ body weight ↓ FAT mass ↑ energy expenditure ↓ fat depots in the liver
Oi-Kano, 2008, [139]	Sprague-Dawley rats ^#^	6–7/6–7	Oleuropein, 1–4 mg/kg, orally	4	↑ UCP1 ^b^ in iBAT	↓ body weight ↓ eWAT and pWAT
Osakabe, 2014, [143]	Wistar rats ^#^	7/8	Flavan-3-ols from cocoa powder, 0.2% w/w, orally	4	↑ UCP1 ^b^ in BAT ↑ UCP3 ^b^ in the SKM	↓ eWAT
Osborn, 2022_I, [144]	C57BL/6 mice ^#^	9–10/9–10	Flavonoid composites, 1% w/w, orally	16	-	↓ body weight and fat mass ↑ lean mass
Osborn, 2022_II, [144]	C57BL/6 mice ^#^	9–10/9–10	4-HPAA 350 µg/day, transdermal	6	↑ PGC-1α ^a^ and UCP1 ^a^ in BAT	↓ fat depots in the liver
Qiao, 2014, [133]	Kunming mice ^#^	8/8	Resveratrol, 200 mg/kg, orally	12	↓ PPAR-γ ^a^, ACC ^a^, FAS ^a^ in eWAT	↓ body weight ↓ visceral and sWAT weight
Tamura, 2020, [145]	C57BL/6J mice ^#^	7/7	Apple polyphenols, 5.0% w/w, orally	4	↑ OXPHOS ^b^ in eWAT ↑ OXPHOS ^b^, PGC-1α ^b^, FGF21 ^a,b^, p-FRS2α ^b^, p-Erk1/2 ^b^, p-MAPK ^b^, UCP1 ^b^ in iWAT	↓ body weight ↓ eWAT, iWAT and BAT weight ↑ thermogenic adaptations
Testai, 2021, [131]	Wistar rats ^#^	6/6	Bergamot juice, 24% v/v, orally	3	↑ PGC-1α ^a^ and UCP1 ^a^ in WAT ↑ plasma levels of irisin	↓ body weight ↓ WAT weight
Wang, 2015, [130]	CD1 mice ^#^	6/6	Resveratrol, 0.1% w/w, orally	4	↑ PRDM16 ^b^, p-AMPK ^b^ and UCP1 ^b^ in iWAT	↓ body weight ↓ iWAT weight and adipocyte size ↑ browning in iWAT ↑ O_2_ consumption in iWAT
Yamashita, 2012, [146]	C57BL6/J mice ^#^	6/6	Cacao liquor procyanidins, 0.5–2.0% w/w, orally	13	↑ p-AMPK ^b^, PGC-1α ^a^ and UCP2 ^a,b^ in WAT ↑ UCP1 ^a,b^ in BAT ↑ UCP3 ^a,b^ in SKM	↓ body weight ↑ lean body mass ↓ total WAT weight ^4^
Zhou, 2018, [142]	C57BL/6J mice ^#^	8/8	EGCG, 1.0% w/w, orally	4	↑ PRDM16 ^a^, PGC-1α ^a^ and UCP1 ^a^ in BAT	↓ body weight ↓ pWAT, eWAT, sWAT ↓ adipocyte size in WAT and lipid droplets in BAT ↑ thermogenic adaptations

Legend: ^#^ Fed a high-fat diet. ^1^ sWAT, eWAT. ^2^ eWAT, pWAT, mesentery WAT, retroperitoneal WAT. ^3^ mesenteric WAT, perinephric WAT, eWAT, iWAT. ^4^ eWAT, mesenteric WAT, retroperitoneal WAT, subcutaneous WAT. ^a^ Gene expression. ^b^ Protein expression. ↑: increased; ↓: reduced.

**Table 3 ijms-24-09276-t003:** Main findings of the in vitro and in vivo studies focused on the involvement of SIRT1 in mediating browning process of polyphenols. Legend: * Compound added at the initial stage of differentiation. Abbreviations: C2C12 murine myoblast cell line; MCP-1, Monocyte Chemoattractant Protein-1; PPAR-γ, peroxisome proliferator-activated receptor γ; UCP1, uncoupling protein 1; mpk, mg/kg/day; mtDNA, mitochondrial DNA; TFAM, mitochondrial transcription factor A. ↑: increased; ↓: reduced.

First Author, Year, Reference	Experimental Model	Concentration or Dose	Compound	Potential Mechanism	Main Effects
Lagouge, 2006, [151]	C2C12 *	50 µM	Resveratrol	↑ SIRT1, PGC-1α, UCP1	↑ oxygen consumption, metabolic consumption
	KKay mice	400 mpk			
	HF-fed C57Bl/6J mice	200, 400 mpk			
Alberdi, 2013, [152]	HF-fed rat	30 mpk	Resveratrol	↑ SIRT1, PGC-1α, UCP1, TFAM, COX2, UCP3 (in muscles)	↑ thermogenesis, whole-body energy dissipation
Li, 2020, [79]	3T3-L1 *	5–20 µM	Resveratrol	↓ PPAR-γ, CEBPα ↑ SIRT1, ATGL, FNDC5, UCP1, PRDM16, CIDEA	↑ Browning-like feature in white adipose tissue
	KKay mice	200 or 400 mpk			
	HF-fed C57Bl/6J mice	200 or 400 mpk			
Andrade, 2014, [153]	HF-fed C57Bl/6J mice	400 mpk	Resveratrol	↑ SIRT1, UCP1, PGC-1α	↑ oxygen consumption ↓ epididymal and retroperitoneal adipose tissue, cholesterol, glucose
Zheng, 2016, [154]	HF-fed SIRT1 (+/−) mice	-	-	↓ UCP1, mtDNA	↓ oxygen consumption and heat production, UCP1, mtDNA ↑ white adipose tissue
Jimenez-Gomez, 2013, [155]	3T3-L1 *	Serum of treated monkeys	Resveratrol	↑ SIRT1, UCP1, PGC-1α	↑ number of adipocytes ↓ inflammation
	HF-fed rhesus monkey	80 or 480 mpk			
Aziz, 2017, [156]	3T3-L1 *	10, 50, or 100 µM	Genistein	↑ SIRT1, UCP1, PGC-1α	↑ Browning-like feature in white adipose tissue, oxygen consumption
Gutiérrez-Salmeán, 2014, [157]	HF-fed mice	1 mpk	Epicatechin	↑ SIRT1, UCP1, PGC-1α	↓ triglycerides, weight gain, cardiometabolic risk factors
Dong, 2014, [158]	HF-fed C57Bl/6J mice	0.1 % w/w	Quercetin	↑ SIRT1/AMPK, PGC-1α, UCP1	↑ Browning-like feature in white adipose tissue ↓ inflammation, body weight gain
Zhao, 2017, [159]	HF-fed rats	120 mpk + 240 mpk	Resveratrol + quercetin	↑ SIRT1/AMPK	↓ inflammation, body weight gain, cholesterol, triglycerides

## Data Availability

Not applicable.

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
