# Peer review of "Influence of Polyphenols on Adipose Tissue: Sirtuins as Pivotal Players in the Browning Process"

_ijms, 2023, doi:10.3390/ijms24119276_

Round 1
Reviewer 1 Report
The aim of the manuscript is challenging; authors are trying to establish, based on literature data the link between polyphenols, adipose tissue disfunction and SIRT activity/expression.
Still, the methodology used in literature search is missing and certain large sections of the manuscript fail to point out the involvement of SIRT in the interplay between adiposity and polyphenols.
Also, the quality of the language should be re-checked and improved
See below specific suggetions
The exact methodology of literature searching should be presented. Use of certain databases or PRISMA (if applies) should be included.
Authors state: “… AT expansion exceed its capability to provide a sufficient an-59 giogenesis resulting in hypoxia, inflammation, oxidative stress, autophagy, apoptosis, fi-60 brosis, and immune cells infiltration.”. Autophagy should not be included in this list but treated separately, since it cannot be considered a destructive factor for the metabolic homeostasis
Authors state “Polyphenols are naturally occurring compounds mainly found in ed-142 ible fruits from the genus Citrus, but they are also present in other food sources, such as 143 red grapes and cocoa. “ – polyphenols are found in many fruits and vegetables not only in this genus, in grapes and cocoa!
The manuscript would benefit from some more citation from the last few years, regarding the interplay of polyphenols and siruins: DOI10.3390/plants11131741, doi: 10.1016/j.pharmthera.2018.03.004, DOI: 10.3390/nu12051344, doi: 10.5483/BMBRep.2019.52.1.290.
Section 5. of the manuscript (Dietary polyphenols and the browning process: evidence from pre-clinical studies) addresses the effects induced by different polyphenols/extracts on the metabolism of adipocytes in animal/cell models but no involvement of SIRT is found. Authors should try to correlate this big section to the aim and the title of the manuscript. Maybe merging sections 5 and 6 could help . Otherwise, a clear justification for this structure should be included in the text
The same observation regarding the lack of SIRT correlation applies to section 7 of the manuscript – lines 523-557 report effects of polyphenols in human adiposity studies without pointing out SIRT correlations.
“Resveratrol, probably one of the most studied/characterized polyphenols” – this is repeated too many times along the manuscript!
Authors should use an uniform manner for citing biblio – either always using the name of the first author (In another study, Alberdi and colleagues) or with numbered references in square brackets. Do not use both methods!
The language quality should be checked by a native speaker in order to improve the manuscript. Some examples that need attention bellow
“adipocytes hypertrophy, that is the increasing in cell size, 49 and adipocytes hyperplasia, which is the increasing in adipocytes number.”
“The stress signal generated by the lack of oxygen 57 promotes angiogenesis and extracellular matrix remodeling but, when the state of over-58 nutrition persists over time, AT expansion exceed its capability to provide a sufficient an-59 giogenesis resulting in hypoxia, inflammation, oxidative stress, autophagy, apoptosis, fi-60 brosis, and immune cells infiltration.”
“SIRT1 is the main responsible…”
“After 12-week, treatment with quercetin 0.1% re-506 duced body weight gain was recorded alongside with reduced insulin resistance and glu-507 cose plasma levels.”???
“SIRT1, UCP1 and PCG-1α activity 509 resulted increased in AT isolated from quercetin fed mice group”???
Reviewer 2 Report
Flori, Piragine et al. reported a comprehensive review of the role of polyphenols in the browning process. BAT in the last decade represent a promising and interesting target for a plethora of metabolic diseases. Therefore, advancing our understanding of the browning process and how it could be modulated by specific molecules or nutrients, represent an intriguing challenge in the lipid metabolism field. The review presented is overall concise and well structured and I recognize the great effort summarizing the metabolic process underlying browning and how polyphenols could affect this pathway. Despite this, I strongly encourage the authors to address the different issues the paper suffers from. Moreover, several issues with the quality of English need to be solved, such as the use of some terms and punctuation. More thorough proofreading before further reviews could be helpful.
Moreover the introduction is not enough insightfully supported by proper references and further details about the rationale behind the use of polyphenols in browning process should be added. A few additional concerns are listed below.
Major points
1. Line 57-60, the interplay between hypoxia and AT expansion needs to be supported by reference.
2. In paragraph 3.1. BAT I would dig more into BAT metabolism and bioenergetics features, such importance of futile cycle/ no shivering thermogenesis and the relevance of BAT in humans. I would consider for instance articles such as “Identification and Importance of Brown Adipose Tissue in Adult Humans” (Aaron M. Cypess, et al, NEJM).
3. The authors stated in line 108 that BAT and BeAT are different in morphology and localization but then, in the following part they described the progenitors of this tissue. Why do the authors not describe the morphological and localization differences?
4. 5. Line 140-141 authors stated “Growing evidence highlighted the ability of polyphenols to promote adipocyte browning and improve metabolic homeostasis.” They should indicate or expand more on this evidence.
5. What is exactly the proposed mechanism in literature on how polyphenols activated or target sirtuins, rather than antioxidant activity per se? There are any data about if the reduced form of polyphenols is most active in targeting sirtuins?
6. Specify what kind of polyphenols the authors are considering in lines 143/144
7. “In recent times, the potential activity of polyphenols in the regulation of lipid metabolism 146 has also attracted particular interest.” (line 146) This sentence needs a reference
8. Pragraph 4.2 is pivotal to introducing the role of sirtuins in the browning process and their use as a metabolic target. This paragraph needs to be expanded and supported with appropriate references.
9. In addition to the graphical abstract in Fig.2, I would add a further image displaying the different SIRT and their metabolic target or mechanism of action.
10. Do the authors know or could find any literature available reporting if a specific diet or beverage/food rich in polyphenols could have possible effects on browning adipose tissue?
Minor points
1. Conclusions should need further details. In light of the reviews and the findings previously reported, do the authors intend that polyphenols could be more effective for weight gain control rather than preventing it?
2. Even though this aspect is behind the main take-home message of the review, the authors should take into account one of the most challenging limitations in the use of polyphenols such as the low bioavailability of most of these compounds. The authors briefly mentioned this key aspect in the conclusions. The authors should provide some more information about possible/ alternative formulations known so far able to improve bioavailability.
3. Concerning paragraph 4 title what does blend exactly stand for? This title is a little bit misleading.
Several issues were met with the use of English (terms used are sometimes misleading). A lack of proper punctuation needs to be addressed and thorough proofreading is mandatory for publication.
Round 2
Reviewer 1 Report
Authors improved the manuscript based on previous suggestions